# Defending Adversaries Using Unsupervised Feature Clustering VAE

**Cheng Zhang**[1]   **Pan Gao**[1]

## Abstract

We propose a modified VAE (variational autoencoder) as a denoiser to remove adversarial perturbations for image classification. Vanilla VAE's purpose is to make latent variables approximating normal distribution, which reduces the latent inter-class distance of data points. Our proposed VAE modifies this problem by adding a latent variable cluster. So the VAE can guarantee inter-class distance of latent variables and learn class-wised features. Our Feature Clustering VAE performs better on removing perturbations and reconstructing the image to defend adversarial attacks.

## 1. Introduction

Adversarial examples become a major challenge for the task of image classification and recognition (Yuan et al., 2019). Several countermeasures have been proposed against adversarial examples, mainly including model-specific hardening strategies and model-agnostic defenses. Typical model-specific solutions like "adversarial training" (Kurakin et al., 2017a; Papernot et al., 2016; Xie et al., 2020; Wong et al., 2020; Shafahi et al., 2019) can rectify the model parameters to mitigate the attacks by using the iterative retraining procedure or modifying the inner architecture. Model-agnostic solutions like input dimensionality reduction or direct lossy image compression (Dziugaite et al., 2016; Das et al., 2017), which attempt to remove adversarial perturbations by input transformations before feeding them into neural network classifiers.

In this paper, we focus on looking for an effective model-agnostic defense strategy (1) can aggressively remove the adversarial perturbations from input images, (2) can reserve sufficient features in input images to ensure the classification accuracy, (3) is still robust to those adversaries which have information about the defense strategy being used. Because the autoencoder are learning to extract useful features and discards useless features to reconstruct the data point. This process gives the autoencoder the potential to remove

the adversarial perturbations. In this paper, we augment the autoencoder with feature clustering. We show that our modified autoencoder can not only defend adversarial attacks but also improve the classification accuracy on clean images.

## 2. Related Work

### 2.1. Autoencoder

Variational autoencder (VAE) (Kingma & Welling, 2014) provides a probabilistic manner for describing an observation in latent space. Thus, rather than building an encoder which outputs a single deterministic value to describe each latent state attribute, VAE formulate the encoder to describe a probability distribution for each latent attribute. With this approach, each latent attribute for a given input will be a probability distribution. When generating output, the encoder is dropped and the decoder randomly samples from latent state distribution to generate a variable as input. Such a randomly sampling process is non-differentiable, so that the training process includes a reparameterization trick. VAE minimizes the negative evidence low bound of $\log p_\theta(x)$:

$$\phi, \psi = \underset{\phi,\psi}{\mathrm{argmax}} \underbrace{\mathbb{E}_{z \sim q_\phi}[\log p_\psi(x|z)]}_{Reconstruction\ Loss} - \underbrace{KL(q_{\phi(z|x)} \parallel p(z))}_{KL\ Regularizer} \quad (1)$$

Different from the generating process of traditional VAE during test, in our work, we preserve the encoder to achieve the goal of removing the perturbation.

### 2.2. Adversarial Attacks

One of the first and simple but quite effective attack is the Fast gradient sign method (**FGSM** (Goodfellow et al., 2015)). It simply takes the sign of the gradient of loss function $J$ (e.g., cross-entropy loss) w.r.t the input image $x$ and multiplies with magnitude $\epsilon$ as perturbations,

$$x_{adv} = x + \epsilon \cdot sign(\bigtriangledown_x L(\theta, x, y)) \quad (2)$$

where $\theta$ is the parameters set of neural network and $y$ is the ground truth label of $x$. The parameter $\epsilon$ is the magnitude of perturbation which controls the similarity of adversarial examples and original image.

By trying to find a high success rate adversarial example but having as small dissimilarity with original image as possible, (Kurakin et al., 2017b) proposed an iterative version of FGSM, **I-FGSM**. It iteratively applies FGSM in

---

[1]Nanjing University of Aeronautics and Astronautics, Nanjing, China. Correspondence to: Pan Gao <PAN.GAO@nuaa.edu.cn>.

*Accepted by the ICML 2021 workshop on A Blessing in Disguise: The Prospects and Perils of Adversarial Machine Learning.* Copyright 2021 by the author(s).

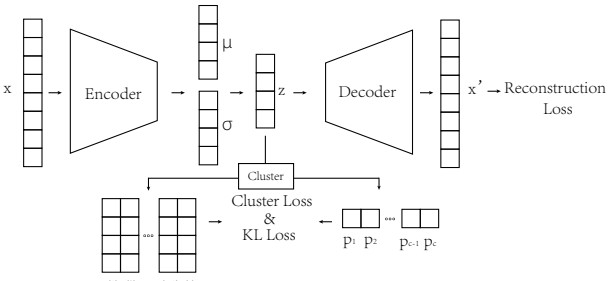

Figure 1. Strucure overview of the proposed Cluster VAE. Where the $c$ is the number of classes, and $\mu(c)$ is the mean of the cluster index $c$, the $p(c)$ is the probability of $z$ belonging to cluster index $c$ classified by our latent classifier.

every iteration and clips the value to ensure per-pixel perturbation below the attack magnitude. **PGD** (Madry et al., 2018) is short for Projected Gradient Descent. Adversarial examples generated by PGD are restarted from different starting point (e.g. by adding tiny random noise to the initial input) for each iteration. **MI-FGSM** (Dong et al., 2018) is called Momentum Iterative Fast Gradient Sign Method which leverages momentum during optimization process. The authors argued that the use of momentum helps to stabilize the update directions for perturbations, and helps to escape from weak local maxima. **BPDA** (Athalye et al., 2018) works by finding a differentiable approximation for non-differentiable pre-processing transformation $g(\cdot)$ or a non-differentiable network layer, possibly via an engineered guess.

## 3. Unsupervised Clustering VAE

Recall the loss function of VAE is simply minimizing the Kullback-Leibler divergence between $Q(z|x)$ and $P(z|x)$, where $Q(z|x)$ is a variational distribution parameterized by the encoder and $P(z|x)$ denotes the true posterior. $Q(z|x)$ is optimized to approximate the $P(z|x)$. Generally, VAE for images generates blurred images, so the VAE is often used as feature extractor. (Ilyas et al., 2019) proposed that adversaries are features, not bugs and there exists robust features and non-robust features in images. That gives us the intuition that we could use a clustering VAE to extract robust features (CVA, 2018; Xie et al., 2016). In addition to the latent coding variables $z$, we add a discrete value $y$ as the cluster index. The cluster index is obtained through training by clustering the features in the latent space, which represents that the representation of the input having the same characteristics is close to each other, benefiting the feature extraction and generation for images of different classes. Thus our Clustering VAE's loss becomes:

$$KL\Big(Q(z,y|x)\Big\|P(z,y|x)\Big) = \mathbb{E}_{z,y\sim Q}\frac{\log Q(z,y|x)}{\log P(z,y|x)} \quad (3)$$

For simplicity, we make two assumptions: (1) the encoder

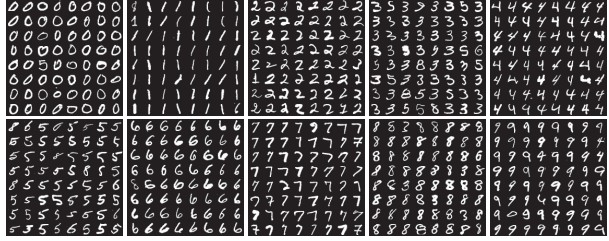

Figure 2. Reconstruction results of clean image clustered by our latent variable classifier.

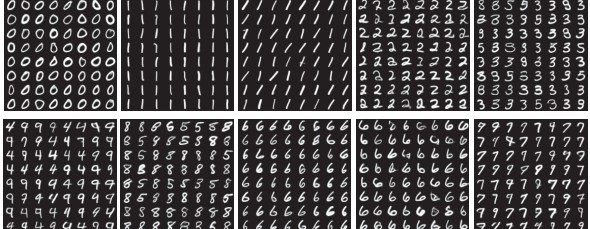

Figure 3. Sampling by means of each cluster by our Cluster VAE.

encodes the input $x$ to latent variables $z$, and clustering the input depends only on $z$; (2) the decoder only uses the latent variable $z$ without the index $y$. This can be mathematically expressed as follows:

$$Q(z,y|x) = Q(y|z)Q(z|x), \quad P(x|z,y) = P(x|z) \quad (4)$$

Applying the Eq. 4 to Eq. 3, with the Bayes rule, we get:

$$KL\Big(Q(z,y|x)\Big\|P(z,y|x)\Big) =$$
$$\mathbb{E}_{z,y\sim Q}\log\frac{Q(y|z)Q(z|x)}{P(x|z)P(z|y)P(y)} + \log P(x) \quad (5)$$

Separating the function inside $\log$, our Clustering VAE loss function will be:

$$Loss = \underbrace{-\log P(x|z)}_{MSE\ loss} + \underbrace{\sum_{y}Q(y|z)\log\frac{Q(z|x)}{P(z|y)}}_{Cluster\ loss}$$
$$+ \underbrace{KL\big(Q(y|z)\big\|P(y)\big)}_{KL\ loss} \quad (6)$$

The first term corresponds to MSE error. The term $Q(y|z)$ can be regarded as a classifier to classify the latent to the corresponding cluster. The $P(z|y)$ in the second term can be seen as a distribution of the latent variable for a cluster index $y$. As in vanilla VAE, $Q(z|x)$ takes the form of normal distribution with mean $\mu(x)$ and standard deviation $\sigma(x)$ that can be learned from data. Here we set the $P(z|y)$ as a normal distribution with mean of $\mu(y)$. Based on our

assumption, we have:

$$Q(z|x) = \sqrt{\frac{1}{(2\pi)^d det(\Sigma(x))}} \exp\left\{-\frac{1}{2}\left\|\frac{z-\mu(x)}{\sigma(x)}\right\|^2\right\}$$

$$P(z|y) = \sqrt{\frac{1}{(2\pi)^{d/2}}} \exp\left\{-\frac{1}{2}\|z-\mu(y)\|^2\right\}$$

$$(7)$$

Because of the reparameterization trick during training process, we know:

$$z = \mu(x) + \Sigma^{\frac{1}{2}}(x) * \epsilon, \epsilon \sim \mathcal{N}(0,1) \quad (8)$$

Substituting the Eq. 8 to Eq. 7 and Eq. 6, the second term of Eq.6 becomes:

$$\sum_y Q(y|z) \cdot [-\frac{1}{2}\log det(\Sigma(x)) - \frac{1}{2}\|z-\mu(y)\|^2] + C \quad (9)$$

where the $C$ is a random value that has nothing to do with the parameters. The second term in Eq. 6 plays the role of making the encoder's clustering of inputs with similar features in the latent space. The third term can be seen as a regularization to make the distribution of each cluster is approximated to prior $P(y)$. We can just set the prior $P(y)$ the uniform distribution. The network structure of our proposed Cluster VAE is shown in Fig. 1.

# 4. Experiment

## 4.1. Clustering Experiment

Firstly, we select the digits from MNIST validation set as the input of encoder, and separate the outputs generated by decoder depending on the cluster results of the latent cluster index. Results are shown in Fig. 2. We can see the clustering results are quite good, except some '3' and '8', '4' and '9' which share similar features. So our Cluster VAE indeed learns to extract the useful features of digits, thus providing the potential for our following work, i.e. removing adversarial perturbations. Note that our clustering training process is different from a basic MNIST classifier. This because that our Cluster VAE is an unsupervised training process, and it forces to extract typical features from training set without the information of labels. Secondly, after the training of our Cluster VAE, we get a normal distribution with mean of $\mu(y)$ for each cluster. We sample the latent variables from each distribution as the input to the decoder and generate outputs using only the decoder. Results are shown in Fig. 3, as we can see, the distribution of each cluster do work to some extent. However, the vertical and slanted '1' are not in same cluster, and '4' and '9' are sharing the similar mean. That is understandable because our Cluster VAE is designed to extract similar features not to classify the digits acting as a classifier. Besides, our purpose is to reconstruct the digits and remove adversarial features, so this will not be an issue.

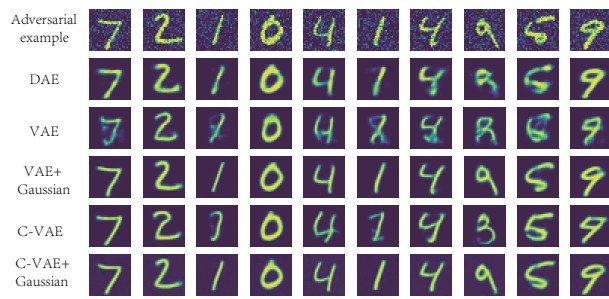

*Figure 4.* Reconstruction of MNIST adversarial examples by different Autoencoders..

## 4.2. Defending Adversaries

### 4.2.1. REMOVING ADVERSARIAL PERTURBATIONS

We introduce the idea of the DAE training process, i.e. adding Gaussian noise in input to mimic the adversarial perturbations, to force autoencoder to extract clean features. That is, we separate the training process into two phrases: first, we train the VAE and Cluster VAE with clean images in training set, and we can get the VAE and C-VAE models. Here we expect the encoder learn to extract robust features from clean image and decoder learn to decodes the robust features to input. Then, we freeze the parameters of the decoder, and train them again with the images added by Gaussian noise, but evaluate the loss between the outputs and clean images. Here, in the second phrase, we expect the encoder learns to extract robust features from noised images. Afterwards, we obtain two trained models, i.e., VAE+G (VAE with Gaussian noised input), CVAE+G (Cluster VAE with Gaussian noised input). Combined with DAE, VAE, and CVAE, we have a total of five models. In the following, we compare the digit results generated for ablation study. The visual reconstructed results are shown in and Fig. 4. When reconstruct from adversarial examples, the visual results of Cluster VAE are better than DAE, and VAE shows the worst results. However, after applying training with Gaussian noised images mentioned above, the VAE and Cluster VAE both show very good visual results.

### 4.2.2. GRAY-BOX ATTACK

In our gray-box setting, the adversary has access to the model architecture and the model parameters, but is unaware of the defense strategy that is being used. We generate adversarial examples using classic and effective attacks mentioned in Section. 2.2. Results are shown in Table. 1. In our experiments, we build a simple CNN classifier for MNIST and adopt ResNet-34 (He et al., 2016) for CIFAR-10 as the model to be attacked. The attack parameters $\epsilon$ of FGSM are 0.2 for MNIST and 0.03 for CIFAR-10. Parameters for other four iterative attacks are $\epsilon = 0.2, stepsize = 0.05$ with 10 iterations for MNIST and $\epsilon = 0.03, stepsize = 0.008$ with 10 iterations for CIFAR-10. As can be seen in the results,

*Table 1.* Summary of model accuracy (in %) for all defenses in Gray-box setting.

| | MNIST | | | | | CIFAR-10 | | | | |
|---|---|---|---|---|---|---|---|---|---|---|
| | None | FGSM | I-FGSM | PGD | MI-FGSM | None | FGSM | I-FGSM | PGD | MI-FGSM |
| No defense | 99.43 | 25.71 | 0.60 | 0.50 | 0.60 | 82.1 | 15.2 | 10.1 | 10.1 | 10.5 |
| DAE | 93.54 | 93.38 | 92.85 | 93.85 | 92.72 | 51.8 | 48.0 | 50.2 | 50.0 | 49.4 |
| VAE | 96.71 | 24.78 | 66.68 | 66.75 | 49.70 | 63.2 | 52.7 | 56.9 | 57.9 | 55.4 |
| VAE+G | 96.27 | 56.33 | 86.34 | 86.82 | 77.92 | 68.6 | 57.4 | 61.8 | 61.5 | 59.2 |
| C-VAE | 97.24 | 84.28 | 90.79 | 91.48 | 85.18 | 79.3 | 48.6 | 58.8 | 58.5 | 53.2 |
| C-VAE+G | 96.72 | 94.93 | 94.93 | 95.82 | 94.43 | 79.2 | 51.6 | 61.2 | 60.4 | 57.4 |

*Table 2.* Accuracy results (in %) in white-box setting.

| | | None | DAE | VAE | VAE+G | C-VAE | C-VAE+G |
|---|---|---|---|---|---|---|---|
| **MNIST** | $Acc_{raw}$ | 99.80 | 96.50 | 97.80 | 97.80 | 98.00 | 97.50 |
| | $Acc_{ae}$ | 0.00 | 1.10 | 19.60 | 28.40 | 33.20 | 37.20 |
| **CIFAR-10** | $Acc_{raw}$ | 82.10 | 51.80 | 63.20 | 68.60 | 79.30 | 79.20 |
| | $Acc_{ae}$ | 0.10 | 0.60 | 13.50 | 27.30 | 30.10 | 35.30 |

*Table 3.* Standard and robust performance(in %) of various adversarial training methods on CIFAR-10 for $\epsilon = 8/255$ and their corresponding training times.

| Methods | VAE | C-VAE | PGD-7 | Free | Fast |
|---|---|---|---|---|---|
| Std acc | 68.6 | 79.2 | 82.46 | 78.38 | 71.14 |
| PGD | 61.5 | 60.4 | 38.86 | 46.18 | 50.69 |
| Time(min) | 14.5 | 22.1 | 68.8 | 20.91 | 7.89 |

our C-VAE training with Gaussian noise can achieve the best accuracy for reconstructing the clean image and best accuracy when defending all the adversarial examples in MNIST dataset. In experiments on CIFAR dataset, our C-VAE can also achieve relatively higher performance in reconstructing the clean image. However, C-VAE shows similar performance when defending adversarial examples with VAE.

### 4.2.3. WHITE-BOX ATTACK

In our white-box setting, the adversary has access to the model architecture and the model parameters, is also aware of the defense strategy that is being used. Since reconstruction using VAE has the random sampling operation which is non-differentiable, and all those five attacks in gray-box setting need to compute the gradients to generate adversarial examples. So we adopt the adaptive attack BPDA (Athalye et al., 2018) in our white-box setting. We implement the evaluation experiment on the released code at GitHub (Athalye et al., 2018), using ResNet-34 (He et al., 2016) as test model, and 1000 iterations and 0.1 learning rate as attack parameters with 1000 images in validation set. The accuracy of various methods on adversarial examples ($Acc_{ae}$) and benign images ($Acc_{raw}$) are reported in Table. 2. Compared to gray-box setting, DAE can defend gray-box attack to large extent while can not defend the adaptive white-box attack. Further, as we can see, all the autoencoders

can hardly defend the BPDA attack, however, our CVAE performs relatively better.

### 4.3. Training Time

Except concerning the defense efficiency of our defense methods, the training time is another indicator we concern. Traditional adversarial training could be extremely time consuming (Madry et al., 2018). Recent works (Shafahi et al., 2019; Wong et al., 2020) proposed several methods to decrease the computational complexity of adversarial training. We compared our training time with those adversarial training, the results of which are shown in Table. 3. Our proposed C-VAE has lower complexity compared to PGD-7 and significantly higher defense efficiency. Compared to Fast and Free adversarial training, our method still exhibits higher efficiency, but brings moderate time increase.

## 5. Conclusion

The experimental results in this paper shows that VAE has the potential to remove the adversarial perturbations and preserve the robust features at the same time. Our proposed Cluster VAE has two merits:

1. Like VAE, Cluster VAE has the random sampling operation which is non-differentiable, which makes the gradient-based adversaries much more difficult to find an adversarial perturbation. Besides, VAE-based defenses have better generalization than adversarial training methods since they are model-agnostic.

2. Our proposed Cluster VAE has the unsupervised clustering operation inside, which can ensure that the encoder extracts robust and label-wised features. Those robust features are important for classification neural networks to make the correct prediction.

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
