# OpenReview forum: "Defending Adversaries Using  Unsupervised Feature Clustering VAE"
_ICML.cc/2021/Workshop/AML — ICML 2021 Workshop AML Poster_

### Official Review · Reviewer_6TE3 · 2021-06-19

**Rating:** Accept
**Confidence:** 4

**Review:**

This paper proposes to remove adversarial perturbations on images by using variational autoencoder (VAE). The proposed method applies a clustering component with a loss on the latent feature of VAE to extract robust features of images and discard non-robust features. The training is performed using Gaussian augmentation of inputs. Experiments under gray-box and white-box attacks show the effectiveness of the proposed method.

Although the idea of using generative models for removing adversarial noise has been explored, this paper proposes a new way to learn a VAE model with feature clustering. The experiments show the effectiveness of the proposed method over baselines.

Below are some detailed suggestions to improve this paper.

1. Obfuscated gradient is a big issue in evaluating adversarial robustness. The authors are encouraged to adopt more diverse attacks to evaluate the robustness of the proposed method, including stronger adaptive attacks, black-box attacks, etc.
2. Some related works [1,2,3] are missing.

[1] Liao et al., Defense against adversarial attacks using high-level representation guided denoiser. CVPR 2018.
[2] Samangouei et al., Protecting classifiers against adversarial at- tacks using generative models. ICLR 2018.
[3] Song et al., Pixeldefend: Leveraging generative models to understand and defend against adversarial examples. ICLR 2018.

---

### Decision · Program_Chairs · 2021-06-21

**Decision:**

Accept (Poster)

**Comment:**

This paper proposed a new approach to remove adversarial perturbations. The paper can be further improve by addressing the reviewer's comments.